# Is Periodontitis a Predictor for an Adverse Outcome in Patients Undergoing Coronary Artery Bypass Grafting? A Pilot Study

**DOI:** 10.3390/jcm10040818

**Published:** 2021-02-17

**Authors:** Stefan Reichert, Susanne Schulz, Lisa Friebe, Michael Kohnert, Julia Grollmitz, Hans-Günter Schaller, Britt Hofmann

**Affiliations:** 1Department of Operative Dentistry and Periodontology, Martin-Luther-University Halle-Wittenberg, 06112 Halle (Saale), Germany; susanne.schulz@medizin.uni-halle.de (S.S.); lisa.friebe@hotmail.de (L.F.); julia.grollmitz@uk-halle.de (J.G.); hans.guenter.schaller@uk-halle.de (H.-G.S.); 2Department of Cardiac Surgery, Mid-German Heart Centre of the University Hospital, 06120 Halle (Saale), Germany; Michael_Kohnert@gmx.de (M.K.); britt.hofmann@uk-halle.de (B.H.)

**Keywords:** periodontitis, adjustment risk, adverse effects, cardiovascular disease, coronary artery bypass surgery, morbidity

## Abstract

Periodontitis is a risk factor for atherosclerosis and coronary vascular disease (CVD). This research evaluated the relationship between periodontal conditions and postoperative outcome in patients who underwent coronary artery bypass grafting (CABG). A total of 101 patients with CVD (age 69 years, 88.1% males) and the necessity of CABG surgery were included. Periodontal diagnosis was made according to the guidelines of the Centers for Disease Control and Prevention (CDC, 2007). Additionally, periodontal epithelial surface area (PESA) and periodontal inflamed surface area (PISA) were determined. Multivariate survival analyses were carried out after a one-year follow-up period with Cox regression. All study subjects suffered from periodontitis (28.7% moderate, 71.3% severe). During the follow-up period, 14 patients (13.9%) experienced a new cardiovascular event (11 with angina pectoris, 2 with cardiac decompensation, and 1 with cardiac death). Severe periodontitis was not significant associated with the incidence of new events (adjusted hazard ratio, HR = 2.6; *p* = 0.199). Other risk factors for new events were pre-existing peripheral arterial disease (adjusted HR = 4.8, *p* = 0.030) and a history of myocardial infarction (HR = 6.1, *p* = 0.002). Periodontitis was not found to be an independent risk factor for the incidence of new cardiovascular events after CABG surgery.

## 1. Introduction

Cardiovascular diseases (CVDs) are the number one cause of death globally. In 2016, CVDs represented 31% of all global deaths [1]. According to the 30th German national heart report published in 2018 [2], coronary heart disease (CHD) with 7.9% and acute myocardial infarction with 5.3% were the leading causes of death in Germany. The treatment of patients with CVD is one of the most common medical tasks in developed industrial countries.

Coronary artery bypass grafting (CABG) surgery is a proven cardiac surgery standard procedure for patients with coronary multivessel disease and/or left main coronary stenosis. CABG survival rates depend amongst other things on age. For instance, in patients <70 years, 4-year adjusted survival rate for CABG was 95.0%, in patients 70 to 79 years of age, survival rate was 87.3%, and in patients ≥80 years, survival was 77.4% [3]. Other independent predictors for mortality after CABG are emergency operation, shock, preoperative renal failure, longer total bypass time, intraoperative stroke, postoperative myocardial infarction, gastrointestinal complications, respiratory failure [4], diabetes [5], and peripheral arterial disease (PAD) [6].

Periodontitis is a chronic multifactorial host mediated inflammatory disease associated with dysbiotic plaque biofilms and characterized by progressive destruction of the tooth-supporting apparatus [7,8]. In a lot of cross-sectional and longitudinal studies, a significant association was demonstrated between periodontitis and CVD [9,10,11,12,13,14] independently of already known risk factors.

Many studies indicate a direct, biologically plausible relationship between periodontitis and CVD. The local host inflammatory response induced by periodontal pathogens promotes the passage of these microorganisms into the blood circulation [15]. Such bacteremia and/or endotoxemia can be caused by invasive dental treatment such as scaling and root planing [16] or even normal daily activities, like tooth brushing, flossing, or food intake [17], and is furthermore associated with the severity of periodontal disease [18]. DNA of key bacteria for periodontitis has been found in both in atheromas [19,20] and heart tissue [21]. In another study, *Aggregatibacter actinomycetemcomitans* was cultured from both specimens taken from periodontal pockets and atheromatous plaque of the same patient [22]. This finding suggests that living bacteria can get from the oral cavity into the coronary arteries and may directly contribute to the pathogenesis of atherosclerosis. In animal experiments, another study showed that *Porphyromonas gingivalis* can invade into heart tissue that has already been damaged by ischemia [23]. This result could indicate that periodontal bacteria not only play a role in the development of atherosclerosis but can also influence the cardiovascular outcome after a primary event.

In a previous study of our group, we investigated the subgingival microbiome in cardiovascular (CV) patients undergoing coronary artery bypass grafting (CABG) in order to identify putative microbial biomarkers for further adverse events after heart surgery. We determined that Saccharibacteria phylum (class: TM7-3, order: CW040, family: F16) was found to be associated with the incidence of secondary CV events (*p* = 0.016) [24]. The present study aims to investigate whether clinical conditions of periodontitis were also associated with the incidence of new cardiovascular events after CABG surgery.

## 2. Materials and Methods

### 2.1. Patients with CVD

Out of a population of 308 patients, 102 patients with CHD for whom an CABG surgery was indicated at the department of cardiac surgery of the Mid-German Heart Centre at the University Hospital Halle (Saale) between January and October 2017 were included in the study. A total of 206 patients could not be included in the study because they did not meet the inclusion criteria, a dental examination before the CABG operation was not possible, a cardiac emergency existed, or the patients did not consent to participation in the study. The most common reason for exclusion was that the subjects had fewer than four own teeth. Follow-up data were generated from 101 patients between January 2018 to June 2019 (dropout rate 0.98%). The study design is summarized in Figure 1. An experienced cardiac surgeon diagnosed patients with CHD and checked the inclusion and exclusion criteria. The following inclusion criteria had to be met: age >18 years, at least 60% stenosis of one of the main coronary arteries demonstrated by angiography, presence of at least four teeth. Exclusion criteria were inability to give written informed consent, subgingival scaling and root planing and/or antibiotic therapy during the last 6 months prior to the examination, pregnancy, and the need for antibiotic prophylaxis against endocarditis according to the criteria of the European Society for Cardiology [25]. Moreover, patients with diseases or disorders such as current drug or alcohol abuse that preclude participation in this clinical study according to investigator judgment were excluded.

### 2.2. Demographic Parameters and Clinical and Cardiological Diagnostics

In order to be able to assess the severity of the CVD, the number of coronaries affected was determined (one-, two- or three-vessel disease). In addition, Canadian Cardiovascular Society (CCS) stages for angina pectoris were determined. Baseline variables such as age, smoking status (never, past, current smokers, and pack years) and current or past diseases (e.g., diabetes mellitus, hypertension, peripheral arterial disease (PAD), and dyslipoproteinemia) were assessed as part of the patient’s medical history. A person who smoked a minimum of one cigarette per day at the time of questioning was considered to be a current smoker. A past smoker had not smoked for at least one year at the time of the survey. The number of pack years was calculated by multiplying the number of packs of cigarettes smoked per day by the number of years of smoking. Furthermore, all patients underwent detailed clinical and biochemical investigation. For instance, intake of drugs such as lipid lowering drugs, oral anticoagulants, and antiarrhythmics was registered. Serum parameters including international normalized ratio (INR) score, hemoglobin (Hb; mmol/L), hematocrit (1/L), creatinine (µmol/L), urea (mmol/L), glycated hemoglobin (HbA1c; mmol/mol), C-reactive protein (CRP; mg/dL), leukocytes (Gpt/L), and platelets (Gpt/L) were recorded.

### 2.3. Dental Anamnesis and Examinations

The dental anamnesis and examination was done one day before the CABG surgery. The patients were asked about the frequency of brushing teeth per day and whether they practiced interdental hygiene using dental floss or interdental brushes. Furthermore, the question was asked whether periodontal therapy in form of scaling and root planing had ever been carried out. In order to be able to estimate an increased occurrence of severe periodontitis within a family, the patients were asked whether there was premature tooth loss due to tooth loosening within relatives of the first degree.

Before the dental examination, the study participants were asked to rinse with an antibacterial mouthwash solution (*Chlorhexamed^®^ FORTE alcohol-free 0.2%, GlaxoSmithKline Consumer Healthcare GmbH & Co. KG, Munich, Germany) in order to reduce the risk of bacteremia due to the probing of dental pockets. The clinical dental assessment involved determining the plaque index (PI) [26] and bleeding on probing (BOP) [27]. In the plaque index, four tooth surfaces were evaluated: mesio-buccal, mid-buccal, disto-bucca, and lingual. In the bleeding index, six sites around each tooth (mesio-buccal, mid-buccal, disto-buccal, disto-oral, mid-oral, and mesio-oral) were examined. BOP was only evaluated after a waiting time of 30 s after probing. Furthermore, the number of decayed, filled, and missing teeth was registered as well as the number of teeth with furcation involvement.

The measurements for both maximal clinical probing depth (PD = distance between gingival margin and the apical stop of the probe) and maximum clinical attachment loss (CAL = distance between the cementoenamel junction and the apical stop of the probe) were taken also at six sites around each tooth. The maximum values for each tooth were taken to calculate the overall mean per participant. In order to obtain reproducible results for BOP, PD, and CAL, the two examiners (L.F. and J.G.) were particularly trained in using a pressure-sensitive calibrated dental probe (UNC 15 0.2 N Aesculap, Tuttlingen, Germany). Particular attention was paid to ensuring that the examiner oriented the probe in the direction of the tooth axis. The reading was made exactly to the millimeter. If one measuring point (gingival margin or cementoenamel junction) was between two markers of the measuring scale, the measurement was estimated to 0.5 mm. For the calibration, both examiners determined PD and CAL twice on five periodontal phantom models (phantom model A-PB, frasaco GmbH, Tettnang, Germany) and on five patients. To assess the reproducibility of the double measurements, the Bland–Altman method was used [28]. The difference (d) of the two measurements was calculated and plotted against the mean of the two measurements. The measurements are sufficiently reproducible if 95% of the differences (d) were in the range d ± 2 × s, where s denotes the standard deviation of the differences. Regarding our two raters, the differences from two measurements for PD and CAL were to 100% in this range d ± 2 × s. Thus, the examiners L.F. and J.G. were able to generate reproducible measurement results.

The clinical periodontitis case definition was held according to the guidelines of the working group of the Centers for Disease Control and Prevention (CDC) [29]. According to the CDC, a severe periodontitis case was diagnosed if at least ≥2 interproximal sites with CAL ≥ 6 mm (not on same tooth) and ≥1 interproximal site with PD ≥ 5 mm were present. A moderate periodontitis case was diagnosed if at least ≥2 interproximal sites with CAL ≥ 4 mm (not on same tooth) or ≥2 interproximal sites with PD ≥ 5 mm (not on one tooth) were present. If no severe or moderate periodontitis was present, periodontitis was diagnosed as mild or absent.

For a more accurate quantification of the root surface affected by attachment loss and quantification of the inflamed epithelial surface, both the periodontal epithelial surface area (PESA) and the periodontal inflamed surface area (PISA) were calculated [30]. For that purpose, a freely downloadable (www.parsprototo.info) Excel spreadsheet was used. In order to calculate PESA, data of CAL and recession were entered. For the calculation of PISA, sites with BOP were recorded additionally.

### 2.4. Follow-Up

Follow-up data were collected from 101 patients. The follow-up was primarily carried out by telephone interview one year after CABG surgery. If follow-up information could not be obtained, we contacted civil registration offices and requested information about current address or date of death. The postoperative outcome was assessed using the major adverse cardiac and cerebrovascular events (MACCE) criteria established for patients with CHD: 1. no event, 2. myocardial infarction, 3. low cardiac output syndrome, 4. ventricular tachycardia (VT), 5. angina pectoris, 6. renewed revascularization surgery, 7. cardiac decompensation, 8. peripheral circulatory failure, 9. stroke/transient ischemic attack (TIA)/prolonged reversible ischemic neurological deficit (PRIND), 10. cardiac death, 11. stroke death, 12. non-cardiac death.

### 2.5. Statistics

Statistical analyses were carried out using commercial software (SPSS v.25.0 package, IBM, Chicago, IL, USA). Values of *p* < 0.05 were considered significant. Categorical variables were documented as number and the corresponding percentage in brackets. For comparisons, the chi-squared test was employed. If the expected values in one group were <5, Fisher’s exact test was performed. Metric demographic, clinical, and serological data were checked for normal distribution using the Kolmogorov–Smirnov test and the Shapiro-Wilk test. As none of the metric values were normally distributed, they were plotted as median and 25th/75th percentiles. For statistical evaluation, the Mann–Whitney U-test was used. For survival evaluation and in order to generate adjusted hazard ratios (HRs), Cox regression was applied. Classic demographic risk factors for CVD such as age, male gender, and nicotine consumption measured as pack years were included in the regression model. Furthermore, the variable periodontitis was integrated into the model according to the CDC classification and in two further models as PESA and PISA. Finally, after bivariate comparisons, significant variables such as PAD and previous MI were included. Although not significant, atrial fibrillation was included because it occurred twice as often in patients with an event compared to patients without an event. Although statistically significant according to bivariate comparisons, the confounders ingestion of oral anticoagulants or antiarrhythmics and early tooth loss among first-degree relatives were not included in the regression model, as the consumption of oral anticoagulants was strictly correlated with prevalence of atrial fibrillation (Pearson correlation coefficient = 0.872, *p* < 0.0001). Antiarrhythmics were only taken by two patients with event, so inclusion in the Cox model was not statistically meaningful. Of all patients, 43.6% could not answer the question about early tooth loss among first-degree relatives, so this variable was also not included in the regression model.

## 3. Results

### 3.1. Incidence of New Cardiovascular Events

No follow-up data could be collected from one patient. Fourteen (13.9%) patients experienced a cardiovascular event within the one-year follow-up after CABG surgery (Figure 1). Eleven patients suffered from angina pectoris, two patients from cardiac decompensation, and one patient died due to a cardiac cause.

### 3.2. Cross-Section Comparisons

There were no significant differences regarding age, sex ratio, BMI, and nicotine consumption between CVD patients with event and without event (Table 1).

Patients with event suffered significantly more frequently from PAD and reported more often a previous myocardial infarction (MI). Moreover, they took anticoagulants and antiarrhythmics significantly more frequently. In addition, there was a trend of an elevated number of CVD patients with atrial fibrillation among the event group. (Table 2).

Regarding dental conditions, patients with new cardiovascular event reported early tooth loss caused by tooth loosening among first-degree relatives (Table 3) significantly more often. However, 43.6% of all study participants could not answer this question. No other dental parameters such as severe periodontitis, PESA, or PISA were significantly associated with the cardiovascular outcome.

### 3.3. Multivariate Survival Analyses

The possible prognostic value of the periodontal parameters CDC, PESA, and PISA were calculated using Cox regression, taking into account the confounders age, gender, pack years, PAD, previous MI, and atrial fibrillation. For severe periodontitis we observed an increased hazard ratio (HR = 2.6) for the combined endpoint, but this result was not significant (*p* = 0.610). In contrast, PAD and previous myocardial infarction were significantly associated with the combined endpoint. Atrial fibrillation also increased the risk for the combined endpoint with borderline significance (Table 4). If instead of CDC the periodontal parameters PESA or PISA were inserted into the regression model, no significance could be achieved for both variables (PESA: HR = 1.001, *p* = 0338; PISA: HR = 1.0, *p* = 0.960).

## 4. Discussion

Periodontitis has a high worldwide prevalence with nearly 40% in the age group of the 35–44-year-olds and nearly 50% in the age group of the 65–74-year-olds [31]. There is increasing evidence that periodontitis can promote the development of systemic diseases such as atherosclerosis and subsequent CVD. Therefore, the aim of our study was to investigate whether periodontitis could influence the cardiovascular outcome after CABG surgery. For periodontitis disease case definition, two different periodontitis classification systems were used in order to compensate disadvantages of a separate system. CDC diagnostics are based on the determination of CAL and PD in interproximal sites. Bleeding upon probing is not taken into account. Three categories of disease, severe, moderate, and no or mild periodontitis, are derived [29]. On the contrary, PESA and PISA are metric variables that can determine the amount of periodontal altered pocket epithelium (PESA) or inflamed pocket epithelium (PISA). It is assumed that in particular the area of the periodontal inflamed tissue may be associated with the systemic inflammatory burden [30].

Among our patients with CVD, the prevalence of severe periodontitis (71.3%, Table 3) was more than two times higher compared to the normal population according to the Fifth German Oral Health Study, which revealed among younger seniors (65 to 74-year-olds) a prevalence of 28.3% [32]. This result may support the importance of severe periodontitis in the pathogenesis of atherosclerosis and CVD. However, we could not show a significant association of severe periodontitis according to CDC definition, PESA, or PISA with the cardiovascular endpoint. For severe periodontitis, only an insignificant trend for the incidence of new cardiovascular events could be shown (Table 4). Thus, we could not demonstrate that severity of periodontitis is a risk factor for a worse postoperative course. The reasons for this result could be manifold. Firstly, it was striking that among our cohort there were no patients in the category of no or mild periodontitis. All study participants had at least moderate periodontitis (Table 3). Therefore, a sharp distinction between periodontally diseased and not diseased was not possible.

The second reason is that both the number of test persons included in our study and the length of the observation period were different in comparison to other studies, which showed a positive association between periodontitis with recurrent cardiovascular events. Dorn et al. [33] investigated 884 survivors of MI and revealed that after an average follow-up of 2.9 years, the mean CAL was associated with recurrent fatal and non-fatal cardiovascular events. Renvert et al. [34] investigated 165 consecutive subjects with acute coronary syndrome (ACS) and 159 medically healthy, matched control subjects regarding periodontal conditions. After an observation period of 3 years, a positive association between alveolar bone loss caused by periodontitis and future ACS events was shown. Since we found at least a trend between severe periodontitis and new cardiovascular events in the present study (Table 4), further studies with more subjects and a longer observation period may be useful.

A third reason may be that other risk factors are more important for the incidence of new cardiovascular events after CABG surgery than periodontitis. In Cox regression, we found a significant positive association between PAD, previous MI, and the cardiovascular endpoint and a trend for atrial fibrillation (Table 4). This result is supported by prior studies. For instance, patients with PAD had poorer long-term survival rates after CABG surgery than patients without PAD [6]. It is assumed that PAD may be a marker of more severe atherosclerosis and subsequent diseases such as CVD. Another possibility is that in spite of successful CABG surgery, the risk of noncardiac mortality may be increased [35]. Another study [36] revealed among patients with PAD a higher incidence of comorbidities in comparison to patients without PAD. Furthermore, a history of MI was associated with increased mortality during the first 30 days after CABG surgery and the incidence of new MI [37]. Other studies have shown a positive association between atrial fibrillation and the outcome of CABG surgery. Preoperative atrial fibrillation was found associated with increased late cardiac morbidity and mortality, poorer long-term survival, higher risk of all-cause mortality, and congestive heart failure [38,39,40].

A further observation should not go unmentioned. Patients with event reported early tooth loss by tooth loosening among first-degree relatives significantly more often compared to subjects without event (Table 3). This result was not checked in the multivariate survival analysis, as 43.6% of patients could not give an answer to the corresponding question during anamnesis. Furthermore, incorrect information from the patient cannot be ruled out. Nevertheless, this observation may indicate the influence of genetic factors in pathogenesis of periodontitis. This is interesting because periodontitis and CVD share common genetic risk factors. For instance, single nucleotide polymorphisms (SNPs) in long non-coding RNA ANRIL (antisense noncoding RNA in the *INK4* locus) were shown to be associated with both CVD and periodontitis [41]. These risk factors may also be associated with a worse outcome after CABG surgery. This hypothesis should also be tested in further studies.

### Limitations of the Study

When interpreting the results of this study, a number of limitations should be noted. First, in comparison to previous studies, the relatively short observation period of one year after CABG has to be mentioned. Secondly, none of the CVD patients included was periodontally healthy. A comparison between CVD patients with and without periodontitis might show a clearer influence of the risk factor periodontitis on the cardiovascular outcome after CABG surgery. However, our patients were aged from 60 to 75 years. In a previous study of our group [42], we determined that among 1002 CVD patients of the same age group, only 2.3% individuals had no periodontitis. Therefore, recruiting CVD patients without periodontitis in the sense of a control group would be very difficult. Thirdly, changes in general conditions, lifestyle habits (in particular smoking status), and possible treatment of periodontitis after CABG might influence the cardiovascular outcome. These factors were not evaluated during follow-up.

## 5. Conclusions

We obtained a severe trend (HR = 2.6) for periodontitis as one predictor for new adverse events within one year after CABG surgery. In order to obtain a significant result, an extension of the observation time would be useful. We confirmed that PAD and previous MI are putative predictors for a poorer outcome after CABG surgery.

## Figures and Tables

**Figure 1 jcm-10-00818-f001:**
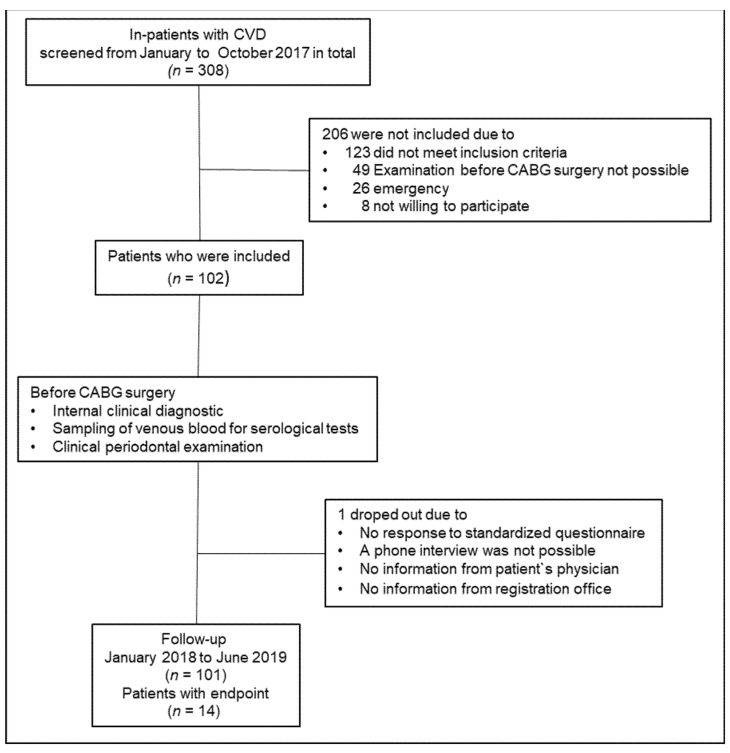
Study design flow chart.

**Table 1 jcm-10-00818-t001:** Demographic variables in dependence of the occurrence of adverse cardiovascular events one year after CABG surgery.

Variable	Entire Study Cohort *n* = 101, Median (25th/75th Percentile) or *n* (%)	Event *n* = 14, Median (25th/75th Percentile) or *n* (%)	No event *n* = 87, Median (25th/75th Percentile) or *n* (%)	*p*-Values
age (years)	69.0 (60.0/75.0)	71.0 (60.8/75.3)	69.0 (60.0/74.0)	0.646 *
females	12 (11.9)	2 (14.3)	10 (11.5)	
males	89 (88.1)	12 (85.7)	77 (88.5)	0.671 ***
BMI (kg/m^2^)	28.7 (25.6/31.0)	29.7 (25.4/31.8)	28.7 (25.6/30.5)	0.476 *
smoking				
current	22 (21.8)	3 (21.4)	19 (21.8)	
past	42 (41.6)	6 (42.9)	36 (41.4)	
never	37 (36.6)	5 (35.7)	32 (36.8)	0.995 **
pack years	7.5 (0/22.5)	3.0 (0/21.3)	8.0 (0/22.5)	0.686 *

BMI, body mass index; * *p* calculated with Mann–Whitney *U*-test; ** *p* calculated with chi^2^-test; *** *p* calculated with Fisher’s exact test.

**Table 2 jcm-10-00818-t002:** Anamnestic and clinical parameters in dependence of the occurrence of adverse cardiovascular events one year after CABG surgery.

Variable	Entire Study Cohort *n* = 101, Median (25th/75th Percentile) or *n* (%)	Event *n* = 14, Median (25th/75th Percentile) or *n* (%)	No event *n* = 87, Median (25th/75th Percentile) or *n* (%)	*p*-Values
Affected coronaries				
One-vessel disease	5 (5.0)	2 (14.3)	3 (3.4)	
Two-vessel disease	21 (20.8)	3 (21.4)	18 (20.7)	
Three-vessel disease	75 (74.3)	9 (64.3)	66 (75.9)	0.214 **
Angina pectoris grade				
CCS 0	32 (31.7)	7 (50)	25 (28.7)	
CCS I	13 (12.9)	1 (7.7)	12 (13.8)	
CCS II	25 (24.8)	2 (14.3)	23 (26.4)	
CCS III	18 (17.8)	2 (14.3)	16 (18.4)	
CCS IV	13 (12.9)	2 (14.3)	11 (12.6)	0.599 **
History of				
Diabetes mellitus	40 (39.6)	7 (50)	33 (37.9)	0.397 ***
Hypertension	88 (87.1)	14 (100)	74 (85.1)	0.205 ***
Dyslipoproteinemia	81 (80.2)	13 (92.9)	68 (78.2)	0.291 ***
PAD	16 (15.8)	5 (35.7) ↑	11 (12.6)	0.044 ***
CVD	39 (38.6)	5 (35.7)	34 (39.1)	1.00 ***
MI	28 (27,7)	8 (57.1) ↑	20 (23.0)	0.020 ***
stroke/TIA	9 (8.9)	0 (0)	9 (10.3)	0.354 ***
Angina pectoris	75 (74.3)	10 (71.4)	65 (74.7)	0.752 ***
PTCA/stent	15 (14.9)	3 (21.4)	12 (13.8)	0.433 ***
Atrial fibrillation	14 (13.9)	4 (28.6)	10 (11.5)	0.102 ***
Blood values				
INR	1.04 (0.99/1.11)	1.04 (0.95/1,12)	1.04 (0.99/1.10)	0.984 *
Hb (mmol/L)	8.8 (8.3/9.4)	8.4 (8.1/9.4)	8.8 (8.3/9.4)	0.437 *
Hematocrit 1/L	0.41 (0.39/0.43)	0.4 (0.38/0.43)	0.41 (0.39/0.43)	0.778 *
Creatinine (µmol/L)	85 (75.5/100)	86.5 (79.5/100.3)	85.0 (74.0/99.0)	0.440 *
Urea (mmol/L)	5.9 (4.5/7.2)	6.8 (5.4/7.6)	5.6 (4.3/6.8)	0.065 *
HbA1C (mmol/mol)	37.6 (31.2/44.4)	39.0 (35.1/52.6)	40.1 (35.9/48.8)	0.769 *
CRP (mg/L)	2.6 (1.2/6.6)	1.4 (0.7/3.9)	2.8 (1.4/6.9)	0.052 *
Leukocytes (Gpt/L)	6.5 (7.6/9.1)	7.2 (5.5/8.5)	7.6 (6.6/9.5)	0.210 *
Platelet (Gpt/L)	238.0 (193.0/269.5)	225.0 (184.0/259.8)	239.0 (193.0/280.0)	0.401 *
Drugs				
Lipid lowering drugs	90 (89.1)	14 (100.0)	76 (87.4)	0.354 ***
Oral anticoagulants	11 (10.9)	4 (28.6) ↑	7 (8.0)	0.044 ***
Antiarrhythmics	2 (2.0)	2 (14.3) ↑	0 (0.0)	0.018 ***

CCS, Canadian Cardiovascular Society; PAD, peripheral arterial disease; CVD, coronary vascular disease; CRP, C-reactive protein; INR, international normalized ratio; Hb, hemoglobin; MI, myocardial infarction; PTCA, percutaneous transluminal coronary angioplasty; TIA, transient ischemic attack; HbA1C, glycated hemoglobin; * *p* calculated with Mann–Whitney *U*-test; ** *p* calculated with chi^2^-test; *** *p* calculated with Fisher’s exact test; **↑** significant differences.

**Table 3 jcm-10-00818-t003:** Dental conditions in dependence of the occurrence of adverse cardiovascular events one year after CABG surgery. Significant differences are indicated with arrows.

Variable	Entire Study Cohort *n* = 101, Median (25th/75th Percentile) or *n* (%)	Event *n* = 14, Median (25th/75th Percentile) or *n* (%)	No event *n* = 87, Median (25th/75th Percentile) or *n* (%)	*p*-Values
Dental anamnesis				
tooth brushing/d				
1	15 (14.9)	1 (7.1)	14 (16.1)	
2	80 (79.2)	12 (85.7)	68 (78.2)	
3	6 (5.9)	1 (7.1)	5 (85.7)	0.678 **
Use of floss/interdental				
brushes				
	29 (28.7)	5 (35.7)	24 (27.6)	0.563 ***
Previous SRP				
	12 (11.9)	3 (21.4)	9 (10.3)	0.366 ***
Early tooth loss among first-degree relatives				
Yes				
No				
Unknown	26 (25.7)	8 (51.7) ↑	18 (20.7)	
Periodontitis (CDC)				
No or mild	0 (0.0)	0 (0.0)	0 (0.0)	
Moderate	29 (28.7)	3 (21.4)	26 (29.9)	
Severe	72 (71.3)	11 (78.6)	61 (70.1)	0.516 **
Plaque index (%)	1.3 (0.9/1.7)	1.2(0.8/1.8)	1.3 (1.0/1.7)	0.293 *
Bleeding index (%)	18.0 (10.1/33.3)	19.0 (13.8/37.0)	17.5 (9.6/33.3)	0.220 *
Pocket depth (mm)	3.0 (2.6/3.6)	2.8 (2.6/3.5)	3.0 (2.6/3.6)	0.738 *
% sites with PD				
<3 mm	34.4 (23.3/52.7)	34.8 (23.2/59.1)	34.4 (23.3/52.6)	0.976 *
3–5 mm	56.7 (45.0/67.9)	64.2 (40.7/70.5)	56.3 (45.2/66.7)	0.353 *
>5 mm	1.7 (0/8.3)	1.2 (0/5.9)	1.7 (0/9.1)	0.373 *
Attachment loss (mm)	3.9 (3.1/4.9)	3.9 (3.1/5.0)	3.9 (3.2/5.0)	0.705 *
% sites with CAL				
<3 mm	16.7 (3.9/32.7)	13.5 (3.5/44.3)	19.3 (4.2/32.1)	0.871 *
3–5 mm	59.4 (47.4/68.5)	60.0 (44.3/76.3)	59.4 (48.3/68.2)	0.596 *
>5 mm	12.7 (3.2/33.3)	11.8 (1.8/35.0)	12.7 (3.2/33.3)	0.735 *
PESA (mm^2^)	1187.8 (831.4/1617.3)	1393.7 (966.2/1778.6)	1165.2 (812.3/1577.9)	0.453 *
PISA (mm^2^)	194.6 (107.6/405.9)	289.7 (164.8/407.9)	191.4 (103.1/419.9)	0.515 *
DMF/T	18 (14.0/22.0)	16.0 (12.0/21.3)	19.0 (14.0/23.0)	0.266 *
Missing teeth	7 (3.0/15.0)	6.0 (2.0/9.8)	7.0 (3.0/17.0)	0.695 *
Teeth with open furcations	0 (0/2.0)	1.0 (0.0/2.0)	0.0 (0.0/2.0)	0.517 *

DMF/T, decayed missing filled/teeth; PD, pocket depth; CAL, clinical attachment loss; PESA, periodontal epithelial surface area; PISA, periodontal inflamed surface area; SRP, scaling and root planing >6 month prior to dental examination, CDC, Centers for Disease Control and Prevention; * *p* calculated with Mann–Whitney U-test; ** *p* calculated with chi^2^-test; *** *p* calculated with Fisher’s exact test; **↑** significant differences.

**Table 4 jcm-10-00818-t004:** Hazard ratios (HRs) for new cardiovascular events adjusted for age, gender, pack years, severe periodontitis according to CDC classification, atrial fibrillation, peripheral artery disease (PAD), and previous myocardial infarction (MI).

Confounding Variables	Hazard Ratio	95% Lower	CI Upper	*p*-Values
Age	0.986	0.912	1.067	0.725
Gender (female)	1.601	0.321	7.976	0.566
Pack years	0.960	0.916	1.005	0.082
Severe periodontitis	2.559	0.610	10.743	0.199
Atrial fibrillation	3.701	0.941	14.562	0.061
PAD	4.836	1.162	20.126	0.030
Previous MI	6.056	1.892	19.379	0.002

PAD, peripheral arterial disease; MI, myocardial infarction; CI, confidence interval.

## Data Availability

The data presented in this study are available on request from the corresponding author.

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
