# Peer review of "Is Periodontitis a Predictor for an Adverse Outcome in Patients Undergoing Coronary Artery Bypass Grafting? A Pilot Study"

_jcm, 2021, doi:10.3390/jcm10040818_

Round 1
Reviewer 1 Report
Is periodontitis a predictor for an adverse outcome in patients undergoing coronary artery bypass grafting (CABG)? – A pilot study
Reichert et al.
Specific comments:
- Abstract: Modify necessity for CABG with who underwent or required CABG
- Line 76 explain in brief why 206 out 308 eligible patients were excluded
- Figure 1 dropped out typo
- Line 94 three-vessel typo
- Line 107 ,] typo
- Line 164 specify N=?
- Line 169 wrong abbreviation VT= Ventricular Tachycardia
- Line 172 Non instead of not
- Line 199 elevated typo
- Line 200 atrial typo
- Table 2 align peripheral arterial dis.
- Line 208 add comma (,) after conditions
- Table 3 align tooth brushing/d
- Table 3 align type of events No or mild
- Table 4 atrial typo
- Please add a limitations section to manuscript including the lack of control group and short follow-up time compared to prior positive studies.
- Is it possible to have CAD without periodontitis…
- Line 278 substitute further by prior
- Line 287 poorer
- Change conclusion to mention severe trend and include that it is after 1 year follow-up after CABG
- Line 302 add 1 year after new adverse events`…
- Line 310 define DG PARO and CP GABA
Author Response
Dear reviewer 1
thank you very much for your invaluable comments while revising our manuscript. We introduced your comments and critical remarks in the revised manuscript. Insertions are highlighted by the "track changes" function.
Best regards,
Stefan Reichert et al.
Step by step answer to the reviewer's comments
Specific comments:
- Abstract: Modify necessity for CABG with who underwent or required CABG
Answer: The definition of the patient has been clarified.
- Line 76 explain in brief why 206 out 308 eligible patients were excluded
Answer: A description was added why 206 patients could not be included.
- Figure 1 dropped out typo
Answer: Figure 1 was not removed because in our view the study design can not be presented so clearly in the text. In the text, however, a better refer to Figure 1 was made.
- Line 94 three-vessel typo
- Line 107 ,] typo
- Line 164 specify N=?
- Line 169 wrong abbreviation VT= Ventricular Tachycardia
- Line 172 Non instead of not
- Line 199 elevated typo
- Line 200 atrial typo
- Table 2 align peripheral arterial dis.
- Line 208 add comma (,) after conditions
- Table 3 align tooth brushing/d
- Table 3 align type of events No or mild
- Table 4 atrial typo
Answer: All corrections and improvements in the layout were carried out.
- Please add a limitations section to manuscript including the lack of control group and short follow-up time compared to prior positive studies.
Answer: A limitations section has been added. The short observation time compared to previous positive studies and the lack of CVD patients without periodontitis were mentioned.
- Is it possible to have CAD without periodontitis…
Answer: Yes. However, our patients aged from 60 to 75 years. In a previous study of our group [Reichert et al. Journal of Clinical Periodontology 2016;43(11):918-925.] we obtained among 1002 CVD patients same age group only 2.3% individuals who had no periodontitis. Therefore, recruiting CVD patients without periodontitis in the sense of a control group would be very difficult.
- Line 278 substitute further by prior
- Line 287 poorer
- Change conclusion to mention severe trend and include that it is after 1 year follow-up after CABG
- Line 302 add 1 year after new adverse events`…
- Line 310 define DG PARO and CP GABA
Answer: All suggestions were taken into account.
Reviewer 2 Report
General comments
This study is a longitudinal clinical study. Its objective is to evaluate the clinical conditions of periodontitis as a predictor of cardiovascular secondary events in patients who have undergone corony artery bypass grafting.
The subject of the associations between periodontal diseases and systemic diseases, and in particular cardiovascular diseases, has been studied for several years. This study is in line with some recent ones on the evaluation of the association between periodontal inflammation or the presence of periodontitis with cardiovascular events and/or mortality (Van Dyke et al., J Periodontol, 2021, DOI: 10.1002/JPER.19-0441 ; Bengtsson et al., Oral Clin Investig, 2021, DOI: 10.1007/s00784-020-03739-x ).
Regarding the organization and description of the work, the authors follow the classic IMRAD plan. The introduction provides sufficient background and is well referenced. The methodology of the periodontal evaluation with two calibrated examiners is rigorous. The inclusion and exclusion criteria at the time of the inclusions (2017) are consistent.
However, a few points need to be clarified, particularly with regard to the Cox regression analysis, and some limitations of the study could be raised. These points are included in the specific comments below.
Specific comments
Materiel and methods
- Page 4-5 lines 164-172: 2.4. Follow up
Patient follow-up appears to have been limited to the occurrence of secondary events following the cardiovascular intervention. Changes in general condition/drugs, lifestyle habits (in particular smoking status), therapeutic management of periodontitis after CABG could also have influenced the occurrence or not of secondary cardiovascular events.
Were the patients questioned on these points during the follow-up period?
If yes, were there any changes compared to the pre-surgical status? Difference in the groups ? These information should be reflected in the results and methods sections.
If not, this should be discussed in the discussion section.
- Page 5 lines 181-183: 2.5 Statistics
Additional information about the construction of the Cox regression model is necessary to help the understanding.
Indeed, how were the variables included in the model chosen?
Were there any preliminary analyses not shown to determine this?
If the model was based on the bivariate analyses (Tables 1, 2, 3) why were items that were significant between groups in the bivariate analyses such as drugs (oral anticoagulants/antiarrhythmics) not included in the cox regression analysis?
Were the data overly correlated?
If so, this should be clarified and discussed.
In regression models, the choice of variables to be included significantly influences the results of the model.
Results
- Page 7: Table 3
In table 3 it is noted "Use of loss/interdental brushes previous SRP". It seems to me that it is floss and not loss. Why is it noted SRP (scaling and root planing)? SRP treatment is one of the exclusion criteria. SRP is also mentioned in the notes at the end of the table.
- Page 7 line 223
« combind » must be replaced by combined
- Page 8 line 231
« atery » must be replaced by artery
Discussion
- Page 8 line 249
For the first time in the manuscript, a "test group" is mentioned. This is disturbing for the reader and needs to be either clarified previously (methods and results) or modified in the discussion.
- Page 8 line 250
« more than two times as high » is difficult to understand. Perhaps « more than two times higher compared to » would be easier to understand.
Author Response
Dear reviewer 2
thank you very much for your invaluable comments while revising our manuscript. We introduced your comments and critical remarks in the revised manuscript. Insertions are highlighted by the "track changes" function.
Best regards,
Stefan Reichert et al.
Step by step answer to the reviewer's comments
Reviewer 2
Specific comments
Materiel and methods
- Page 4-5 lines 164-172: 2.4. Follow up
Patient follow-up appears to have been limited to the occurrence of secondary events following the cardiovascular intervention. Changes in general condition/drugs, lifestyle habits (in particular smoking status), therapeutic management of periodontitis after CABG could also have influenced the occurrence or not of secondary cardiovascular events.
Were the patients questioned on these points during the follow-up period?
If yes, were there any changes compared to the pre-surgical status? Difference in the groups ? These information should be reflected in the results and methods sections.
If not, this should be discussed in the discussion section.
Answer: We fully agree with the reviewer that changes in lifestyle, drug consumption or possible periodontal therapy could have an impact on the incidence of recurrent events. However, these data were not collected in the follow-up. This is why this important point has been included in the "limitations of the study" section.
- Page 5 lines 181-183: 2.5 Statistics
Additional information about the construction of the Cox regression model is necessary to help the understanding.
Indeed, how were the variables included in the model chosen?
Were there any preliminary analyses not shown to determine this?
If the model was based on the bivariate analyses (Tables 1, 2, 3) why were items that were significant between groups in the bivariate analyses such as drugs (oral anticoagulants/antiarrhythmics) not included in the cox regression analysis?
Were the data overly correlated?
If so, this should be clarified and discussed.
Answer: The statistics section has been expanded as follows: For survival evaluation and in order to generate adjusted Hazard ratios (HR) Cox regression was applied. Classic demographic risk factors for CVD such as age, male gender and nicotine consumption measured as pack years were included in the regression model. Furthermore, the variable periodontitis was integrated into the model according to the CDC classification and in two further models as PESA and PISA. Finally, after bivariate comparisons, significant variables such as PAD and previous MI (see Table 2) were included. Although not significant atrial fibrillation was included because it occurred twice as often in patients with an event compared to patients without an event. Although statistically significant according to bivariate comparisons, the confounders ingestion of oral anticoagulants or antiarrhythmics (Table 2) and early tooth loss among related 1st degree (Table 3) were not included in the regression model. Reason: The consumption of oral anticoagulants was strictly correlated with prevalence of an atrial fibrillation (Perason correlation coefficient = 0.872, p <0 .0001). Antiarrhythmics were only taken by two patients with event, so that inclusion in the cox model was not statistically meaningful. 43.6% of all patients could not answer the question about early tooth loss among related 1st degree so that this variable was also not included in the regression model.
In regression models, the choice of variables to be included significantly influences the results of the model.
Answer: We agree with this statement. So we tried to include only key confounders and limit their numbers.
Results
- Page 7: Table 3
In table 3 it is noted "Use of loss/interdental brushes previous SRP". It seems to me that it is floss and not loss. Why is it noted SRP (scaling and root planing)? SRP treatment is one of the exclusion criteria. SRP is also mentioned in the notes at the end of the table.
Answer: Use of floss is correct. SRP 6 months before the basic examination was an exclusion criterion. Here the question was asked about SRP> 6 months before the basic examination. This has now been noted in the footnote of table 3.
- Page 7 line 223
« combind » must be replaced by combined
- Page 8 line 231
« atery » must be replaced by artery
Answer: All corrections were carried out.
Discussion
- Page 8 line 249
For the first time in the manuscript, a "test group" is mentioned. This is disturbing for the reader and needs to be either clarified previously (methods and results) or modified in the discussion.
Answer: The term test group has been deleted and replaced by patients with CVD
- Page 8 line 250
« more than two times as high » is difficult to understand. Perhaps « more than two times higher compared to » would be easier to understand.
Answer: This sentence has been improved according to this suggestion.
Round 2
Reviewer 2 Report
The authors responded to all comments. The additions regarding the methodology and limitations of the study are satisfactory.